# Selections that isolate recombinant mitochondrial genomes in animals

Hansong Ma, Patrick H O'Farrell*

Department of Biochemistry and Biophysics, University of California, San Francisco, San Francisco, United States

**Abstract** Homologous recombination is widespread and catalyzes evolution. Nonetheless, its existence in animal mitochondrial DNA is questioned. We designed selections for recombination between co-resident mitochondrial genomes in various heteroplasmic *Drosophila* lines. In four experimental settings, recombinant genomes became the sole or dominant genome in the progeny. Thus, selection uncovers occurrence of homologous recombination in *Drosophila* mtDNA and documents its functional benefit. Double-strand breaks enhanced recombination in the germline and revealed somatic recombination. When the recombination partner was a diverged *Drosophila melanogaster* genome or a genome from a different species such as *Drosophila yakuba*, sequencing revealed long continuous stretches of exchange. In addition, the distribution of sequence polymorphisms in recombinants allowed us to map a selected trait to a particular region in the *Drosophila* mitochondrial genome. Thus, recombination can be harnessed to dissect function and evolution of mitochondrial genome.

## Introduction

Homologous recombination operates in organisms from bacteriophage to human. This includes the mitochondrial genomes in many plant and fungal species (*Rank and Bech-Hansen, 1972*; *Dujon et al., 1974*; *André et al., 1992*; *Zassenhaus and Denniger, 1994*; *Shedge et al., 2007*). Nonetheless, there is very little support for recombination in animal mitochondria (*Elson et al., 2001*; *Berlin et al., 2004*; *Hagström et al., 2013*); lack of an identified mitochondrial RecA homolog, evidence of continuous lineages of mitochondrial haplotypes and a failure to detect recombinants in propagated heteroplasmic lines are taken as indications that it does not occur.

Despite evidence arguing against recombination of animal mitochondrial genomes, a variety of exceptional reports suggest that it can occur. The remarkable 'double uniparental inheritance' pattern of mitochondrial genomes in some bivalve mollusks has been associated with rare recombination events on at least an evolutionary time scale (*Ladoukakis and Zouros, 2001*; *Ladoukakis et al., 2011*). One human patient was reported to carry recombinant genomes (*Kraytsberg, 2004*), and there have been reports of recombinant mitochondrial genotypes in some species like lizard and fish (*Guo et al., 2006*; *Ciborowski et al., 2007*; *Ujvari et al., 2007*), but these reports are based on single individuals without documentation of parents or origin of the genomes presumed to have recombined. Given opposing observations, such as the introgression of intact genomes from one species into another (*Solignac, 2004*), it is not clear whether the cases reported are exceptional, or whether we have simply lacked the experimental power to directly demonstrate recombination in animals.

Recombination between small regions of nonallelic homology has been proposed to underlie deletion and insertion mutations (*Mita et al., 1990*; *Bacman et al., 2009*; *Fukui and Moraes, 2009*). However, when precise, recombination can generate favorable combinations of alleles, which, when coupled to the action of purifying selection, could increase in abundance to restore function

*For correspondence: ofarrell@cgl.ucsf.edu

**Competing interests:** The authors declare that no competing interests exist.

**eLife digest** Animals store the main part of their DNA—including all of the genes that are required to build and maintain an individual—inside their cells in a structure called the nucleus. Most of the information stored in the DNA is stored in duplicate, with one copy inherited from the individual's mother via the egg, and the other from the father via the sperm. This duplicate storage allows a very important damage repair process to occur, where undamaged sequences in one copy can be used to repair damage in the other. This process of homologous repair uses mechanisms that are also used in another important genetic process. When sperm and egg cells are formed, the parental DNA goes through a process called homologous recombination, in which DNA molecules are cut and reassembled into new arrangements. This recombination process 'shuffles' genetic combinations, making every individual unique—a process of great evolutionary importance that allows natural selection to act on distinct traits.

The structures inside cells that generate energy—called mitochondria—also contain DNA, which is inherited only from mothers. Little is known about whether recombination is possible in the mitochondrial DNA of animals.

Ma and O'Farrell used genetics techniques to investigate recombination in the mitochondria of fruit flies. One experiment tracked how a mutation that makes flies less healthy at high temperatures spread as flies were bred for several generations. When the mutation was associated with a mitochondrial genome that had a strong drive towards replication, the mutation became more widespread over time, and in most cases, this eventually resulted in the mutation killing the flies. In rare cases, however, a few flies survived, giving rise to a healthy population. Molecular analyses revealed that, in these survivors, the defective genome had recombined with the other mitochondrial DNA to produce a new genome that lacked the mutation but retained the high replicative drive. This new recombinant genome worked normally and was able to resist the spread of the defective genomes.

In addition, by artificially cutting mitochondrial DNA, Ma and O'Farrell show that such 'double-strand breaks' lead to recombination, signaling a role for homologous repair in the repair of damaged, in this case broken, DNA. Recombination is also possible between the mitochondrial DNA of two different fruit fly species, and this recombination process can assemble long stretches of DNA.

Now that the recombination of animal mitochondrial DNA is known to be possible, future work will be required to understand how it works and how it affects evolution.

(*Muller, 1932*). These positive or negative impacts on gene function could influence evolution, the behavior of disease mutations, and the age-associated degenerative changes of the mitochondrial genome.

Until recently, several factors have hindered detection of recombination of mitochondrial sequences. Chief among these, uniparental inheritance largely limits exposure of mitochondrial genomes to sibling genomes differing only at newly mutated sites (*Birky, 1995*; *DeLuca and O'Farrell, 2012*; *Sato and Sato, 2013*). Additionally, rare recombinant genomes can be difficult to detect: they can be stochastically lost during the random segregation, and if transmitted, they can be hard to track amid the chaotically segregating genomes. Finally, there are few markers suitable for design of conditions that would select for a rare recombinant genome.

Previous work in *Drosophila* showed that germline expression of a restriction enzyme targeted to mitochondria results in potent selection against mitochondrial genomes carrying a cognate cleavage site (*Xu et al., 2008*). Using this selection, a number of variant genomes lacking a particular site have been selected. In addition to removing a restriction site, these selected changes often also alter an encoded gene product (*Figure 1A*). One of the variants that lost a XhoI site is a temperature-sensitive lethal mutation of *mt:CoI* that can be counter-selected at high temperature (*Hill et al., 2014*; *Ma et al., 2014*). Moreover, one can transfer cytoplasm between early *Drosophila* embryos of different mitochondrial genotypes to create heteroplasmic lines that carry both the recipient and donor genomes for multiple generations (*Matsuura et al., 1989*; *Ma et al., 2014*). In addition to these tools, characterization of diverged mitochondrial genomes has revealed marked differences in their

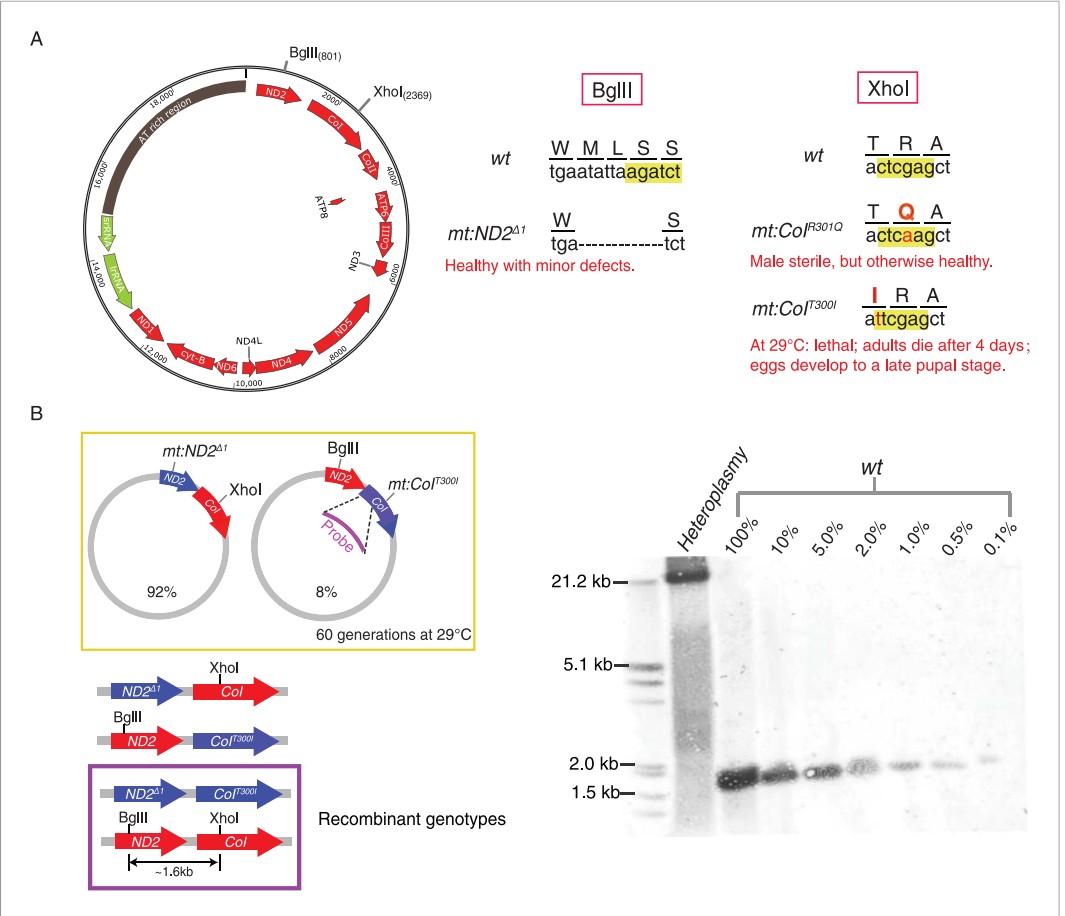

**Figure 1**. Without selection, no recombination was detected in a stable heteroplasmic line after 60 generations. (**A**) Mutants at the BglII and XhoI sites of mtDNA used in this study (*Xu et al., 2008*; *Ma et al., 2014*). The mitochondrial genome of *Drosophila* resembles that of mammals. It has little intergenic spacing, and encodes 13 polypeptides, all of which are involved in oxidative phosphorylation as well as 22 tRNAs and two rRNAs required for mtDNA translation. A single non-coding region (~5 kb) called the 'AT-rich region' (dark brown) as it contains >90% A and T residues includes origins of replication and some repeated sequences of unknown function (*Lewis et al., 1994*). The genome contains one BglII and one XhoI site in the coding regions of *mt: ND2* and *mt:CoI*, respectively. (**B**) No wild-type genome was detected by Southern blotting analysis in heteroplasmic lines where *mt:ND2^{del1}* and *mt:CoI^{T300I}* were maintained in the same population for more than 60 generations at 29°C. In the heteroplasmic lane, 40 adults were sacrificed and their mtDNA were cut with both BglII and XhoI, and probed by a DIG-labeled sequence that hybridizes to mt1579–mt2369. The sensitivity of the Southern analysis was measured by loading a series of dilutions of wild-type mtDNA cut with both enzymes from 40 adult flies.

abilities to compete for transmission when combined in heteroplasmic combinations, a feature that we have been able to use as another selectable trait. In this work, we have combined these tools to create a powerful system in which we can test for the existence of homologous recombination and select for recombinant genomes.

Here, we provide clear evidence for homologous genetic exchange between *Drosophila* mitochondrial genomes under various conditions. The complete genomic sequence of parental and recombinant molecules details exchange events, and several recombinants are shown to involve transfer of a substantial segment of sequence from one genome to the other. We also show that exchange is stimulated by double-strand breaks (DSBs), as is recombination in many systems. Importantly, the success of the selections that we have applied shows that production of favorable combinations of alleles by recombination, even if rare, can have a profound benefit.

## Results

### Direct screening for recombinant genomes in a heteroplasmic line

We used Southern analysis to test for recombination between genomes distinguished by differences in restriction sites. A line heteroplasmic for *mt:ND2^{del1}*, which lacks the BglII site, and *mt:CoI^{T300I}*, which lacks the XhoI site, carries both genomes stably at 29°C (*Figure 1B*) (*Ma et al., 2014*). Each of the genomes of this heteroplasmic line has a defect complemented by the other so that the line is healthy but neither genomes is lost, a balancing selection (*Ma et al., 2014*). Recombination should produce wild-type genomes distinguished by the presence of both restriction sites and consequent production of a ~1.6-kb fragment upon cutting with both BglII and XhoI. Even after maintenance of this line for more than 60 generations, this 1.6 kb band was not detected (*Figure 1B*). Reconstruction shows that this Southern assay can detect recombinant molecules at the level of 1 in 1000 (*Figure 1B*). Thus, like related experiments by others (*Hagström et al., 2013*), this physical assay failed to detect recombination. We conclude that recombination in this situation is not frequent (*Sato et al., 2005*).

### Selection reveals recombination between mitochondrial genomes

We then developed methods to genetically select for recombinant mitochondrial genomes in the hopes that such approaches would both detect its occurrence and allow isolation of the recombination product. To give recombinant genomes an advantage, we produced a heteroplasmic line wherein one genome was compromised by a temperature-sensitive mutation and the other was compromised in its ability to compete for transmission. A temperature-sensitive genome with two mutations, *mt:ND2^{del1}* + *mt:CoI^{T300I}*, was introduced into a strain with a genome named *ATP6[1]* (*Celotto et al., 2006*, *2011*). High temperature, 29°C, selected against the *mt:CoI^{T300I}* allele, and in previously analyzed heteroplasmic lines where the partner genome was a closely related wild-type genome, this selection resulted in a multi-generational decline of the temperature-sensitive genome and its eventual elimination (*Ma et al., 2014*). To our surprise, when in competition with the *ATP6[1]* genome, which is distinguished by numerous sequence polymorphisms and a shorter AT-rich region, the temperature-sensitive genome displaced the *ATP6[1]* genome over a few generations at 29°C, even though the flies homoplasmic for *ATP6[1]* are relatively healthy and apparently more robust than *mt:ND2^{del1}* + *mt:CoI^{T300I}* flies at both temperatures. The decline of *ATP6[1]* leaves the temperature-sensitive genome without complementing *mt:CoI* activity, and the entire population dies after several generations (*Figure 2A*).

To select for possible recombinant genomes, we followed five independently established lines with a high starting proportion of *ATP6[1]* genomes (50–90%) at 29°C. Initially, all the lines were healthy and grew productively, but with decline in *ATP6[1]* abundance, the health of the lines held at 29°C declined abruptly after five or six generations and died out within the next few generations. One line went through a similar crisis with diminished survival, recovered and continued to produce viable progeny in subsequent generations. Southern analysis showed emergence of a new genotype, which was cut by XhoI (like the *ATP6[1]* genome), but had a long AT-rich region (like the double mutant) (*Figure 2B*). To map recombination sites, we sequenced the parental and recombinant genomes using the PacBio single molecule real-time sequencing technique (SMRT), whose long reads gave us unambiguous sequence that included the repeats of the 5 kb highly AT-rich noncoding region (*Figure 1A* and *Figure 2*). Complete sequences of these genomes revealed that the two parental genomes differed by more than 100 SNPs plus >20 indels, and the *ATP6[1]* genome lacked ~1.6 kb of the AT-rich region that was present in the *mt:ND2^{del1}* + *mt:CoI^{T300I}* genome (*Figure 2C* and *Figure 2—figure supplement 1A*). The recombinant genome was the result of an exchange of a large continuous segment to produce an ~60%/40% chimera of the *ATP6[1]* and *mt:ND2^{del1}* + *mt:CoI^{T300I}* genomes that lacks the *mt:ND2* and *mt:CoI* mutations but contains the non-coding region of the *mt:ND2^{del1}* + *mt:CoI^{T300I}* genome (*Figure 2C*).

Southern analysis also showed the abundance of the recombinant genome at later stages of the selection. After generation 6, the original *ATP6[1]* genome was no longer detected (*Figure 2B*), and a new heteroplasmic line was formed with the other parental genome *mt:ND2^{del1}* + *mt:CoI^{T300I}* and the recombinant genome. This line was viable at 29°C as the temperature-sensitive defect of the double mutant is complemented by the *ATP6[1]* *mt:CoI* of the recombinant genome (*Figure 2D*). Over subsequent generations, a multigenerational selection for function caused an increase in the

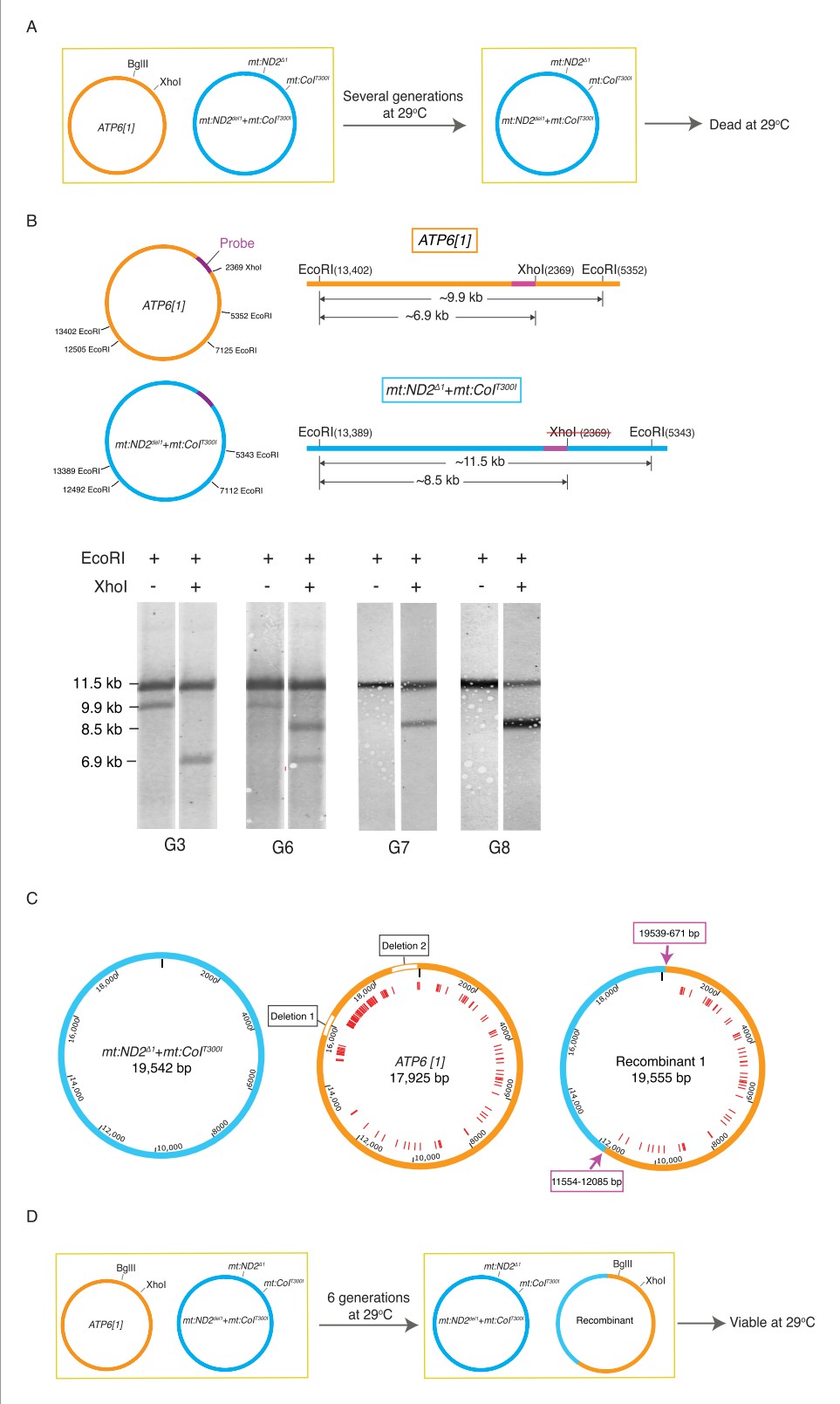

**Figure 2**. Selection revealed homologous recombination in a heteroplasmic line containing the *ATP6[1]* genome and the temperature-sensitive double-mutant: *mt:ND2del1* + *mt:CoIT300I*. (**A**) The abundance the *ATP6[1]* genome declined when co-resident with *mt:ND2del1* + *mt:CoIT300I* at 29°C. After several generations, the flies at 29°C started to
*Figure 2. continued on next page*

*Figure 2. Continued*

die. (**B**) A combination of a restriction fragment length polymorphism and a restriction site difference reveals the emergence of a recombinant genome. mtDNA isolated from 40 adults from each generation was cut with EcoRI in the presence or absence of XhoI. The schematics show the distribution of the EcoRI and XhoI sites on the whole parental genomes (left) and a detail of the largest EcoRI fragment with the position (purple bar) of a hybridization probe (center). Southern analysis shows single and double cut samples taken at different generations during the selection. Only the two parental bands were detected early, from G0 to G5 (shown for G3). From G6 onward, a third EcoRI fragment appeared that had a length characteristic of the $mt:ND2^{del1}$ + $mt:CoI^{T300I}$ genome but with a XhoI site. By G7, the $ATP6[1]$ specific fragment was not detected while a new apparently recombinant genome dominated the population. (**C**) Detailed maps of three genomes sequenced by PacBio SMRT. Red lines indicate mismatches between $ATP6[1]$ and $mt:ND2^{del1}$ + $mt:CoI^{T300I}$ sequences. The $ATP6[1]$ genome also lacks ~1.6 kb of the AT-rich region (two type I repeats and two type II repeats, see **Figure 2—figure supplement 1A** for details). Pink arrows indicate approximate points of exchange with the given range defined by the nearest neighboring polymorphisms (see deposited full sequences in GenBank as KT174472, KT174473 and KT174474). (**D**) Proposed progression leading to the recombinant. The original $ATP6[1]$ genome is still displaced, but the newly emerged recombinant competes effectively and persists. By generation 6, the abundance of the recombinant genome is sufficient to complement the temperature-sensitive genome so that some viable flies sustain the line. Over subsequent generations at 29°C, the recombinant genome increases in relative abundance because of selection against the temperature-sensitive genome.

The following figure supplement is available for figure 2:

**Figure supplement 1**. mtDNA maps for the two parental genotypes ($ATP6[1]$, $mt:ND2^{del1}$ + $mt:CoI^{T300I}$) and two other recombinants.

proportion of the recombinant genome (**Figure 2B**), showing that it had lost the transmission disadvantage of the parental $ATP6[1]$ genome. This led us to conclude that the ability to compete for transmission is localized to the sequences of the recombinant that came from the temperature-sensitive genome. This includes the entire AT-rich regulatory region and some flanking sequences distinguished by three SNPs (**Figure 2C**).

We later isolated two other recombinant genomes by following another 46 heteroplasmic lines as above. Both recombinants had a size (~19.5 kb) similar to the temperature-sensitive genome, implying that $mt:ND2^{del1}$ + $mt:CoI^{T300I}$ was the source of the regulatory region. By sequencing the coding region, we show that one recombinant has the entire coding sequence of the $ATP6[1]$ genome, whereas the other contains a much smaller segment of $ATP6[1]$ extending at least from mt671 to mt5978, with the rest of the coding region belonging to the $mt:ND2^{del1}$ + $mt:CoI^{T300I}$ genome (**Figure 2—figure supplement 1B**). We conclude that our selection has isolated recombinant genomes including extensive stretches of sequence originating from the parental genomes and exhibiting functional traits of these parental genomes.

## Homologous exchange upon introduction of reciprocal DSBs

Normal meiotic recombination is induced by DSBs and studies in many contexts have suggested that DSBs, whether experimentally produced or secondary to DNA damage, greatly stimulates recombination. Indeed, DSBs are thought to be the key initiators of homologous exchange. To test the importance of DSBs for mtDNA recombination, we set up a condition to select for recombination in conjunction with restriction cutting of the mtDNA.

We had produced genomes that lacked the XhoI site, $mt:CoI^{R301Q}$, or the BglII site, $mt:ND2^{del1}$. While each genome is resistant to one enzyme, expression of mito-BglII and mito-XhoI simultaneously in the germline should cut either of these genomes. Indeed, germline expression of both enzymes in flies that were homoplasmic for either $mt:ND2^{del1}$ or $mt:CoI^{R301Q}$ led to sterility in females (**Figure 3A**). As previously shown, resistance ought to emerge at a low frequency due to mutations at the second restriction site (**Xu et al., 2008**). Indeed, upon selection ~1% of the females were weakly fertile, giving a few escaper F1 progeny harboring a variety of mutant alleles that inactivate the remaining restriction site. In contrast, even though the combination of enzymes was able to select against both genotypes, when flies were heteroplasmic for the two genomes (in a 50:50 ratio), fertile females were very

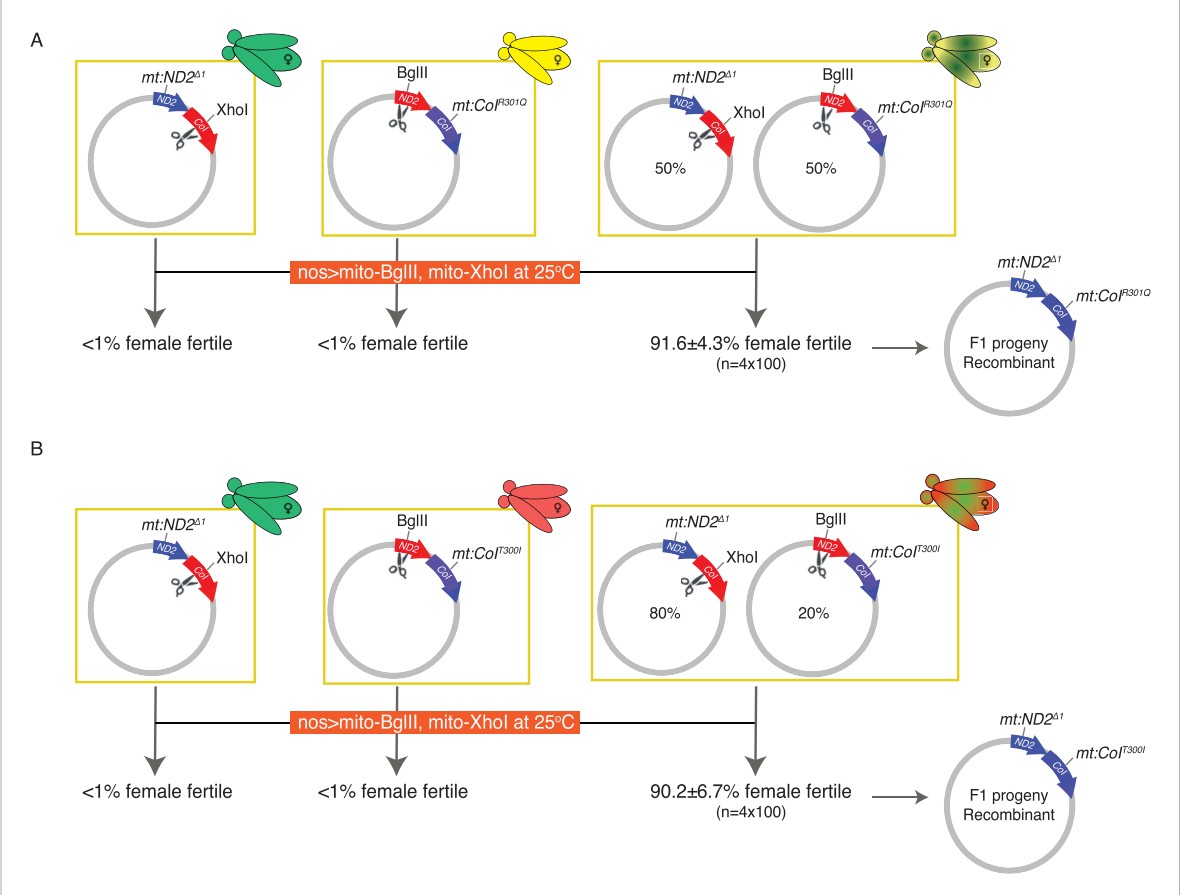

**Figure 3**. Introducing DSBs into both parental genomes vastly induces homologous recombination in two heteroplasmic lines. (**A**) Expression of both mito-BglII and mito-XhoI enzymes in the germline effectively sterilizes most females carrying either $mt:ND2^{del1}$ or $mt:CoI^{R301Q}$ genome, resistant to only one of the enzymes, but resistant progeny appear at a much higher frequency if the female carries both types of sensitive genomes. The resistant progeny from the heteroplasmic parent are homoplasmic for a newly generated recombinant genotype carrying both parental alleles: $mt:ND2^{del1} + mt:CoI^{R301Q}$. (**B**) The same level of rescue was observed when a different heteroplasmic line was used, which had a different starting ratio of two parental genomes. Results are means ±SD (n = 4 × 100 for each heteroplasmic line).

frequent (>90%), and most (60%) were as fecund as wild type. PCR, DNA sequencing, and phenotypic analysis showed that these progeny carried a single genome with the restriction-resistant alleles found in the two parents (*Figure 3A*).

A second heteroplasmic line where the $mt:CoI^{R301Q}$ allele was replaced with the $mt:CoI^{T300I}$ allele gave a similar frequency of recombinant progeny, even though the starting ratio for the two parental genomes was not equal (*Figure 3B*). The resulting recombinant lines exhibited the phenotypes expected for the input alleles. For instance, $mt:ND2^{del1} + mt:CoI^{R301Q}$ flies were male sterile, and $mt:ND2^{del1} + mt:CoI^{T300I}$ flies were temperature lethal.

The success of a second and very different selection confirms that homologous exchange between mitochondrial genomes can occur. Most females exposed to the restriction enzyme selection were fertile, while in the selection without DSBs (above, *Figure 2*), the recombinant emerged from a population over several generations of selection. The difference suggests that the frequency of recombination upon expression of the two restriction enzymes is much higher. We propose that DSBs produced by restriction enzyme cutting induces exchange as well as selecting for the products that are resistant to cutting.

To test whether recombination could also occur in somatic tissues, we examined the consequence of expressing both restriction enzymes in the developing eye under the control of ey:GAL4 driver. The high level of expression that occurs at 29°C results in pupal lethality (headless phenotype) in 99% flies

that are homoplasmic for mitochondrial genomes mutant at only one site and the few survivors are eyeless or have a small eye phenotype. Similar expression in flies heteroplasmic for the two mutant genomes gave survivors at 10% with most survivors showing well-developed eyes (*Supplementary file 1*), suggesting that recombination of mtDNA also occurred in somatic tissues exposed to the double restriction enzyme selection. Somatically, active recombination could impact the stability of the mitochondrial genome in the soma, a factor thought to be important in aging (*Cortopassi et al., 1992*; *Liu et al., 1998*; *Cao et al., 2001*; *Bender et al., 2006*).

## Recombination upon introduction of a single DSB

Given the high frequency of recovered recombinants when both genomes of a heteroplasmic strain were cut, we asked whether we could detect recombination if DSBs were introduced into only one genome. To do this, we made a heteroplasmic line containing the wild-type genome and the temperature-sensitive double-mutant $mt:ND2^{del1}$ + $mt:CoI^{T300I}$, and used expression of restriction enzyme to select against wild-type genome and used the temperature-dependent selection against the double mutant. When we expressed mito-BglII in the germline while keeping the flies at 29°C, about 14.6% of the F1 females were viable and fertile (*Figure 4A*). PCR analysis and restriction digestion showed that the progeny had lost the wild-type allele of $mt:ND2$, which was targeted by the restriction enzyme, but they were heteroplasmic for a new $mt:ND2^{del1}$ genome and the double-mutant parental $mt:ND2^{del1}$ + $mt:CoI^{T300I}$ genome (*Figure 4A*). Apparently, the wild-type mtDNA was cut and homologous repair or exchange using the mutant $mt:ND2^{del1}$ sequence led to loss of the site. Initially, the abundance of the newly generated $mt:ND2^{del1}$ genome was very low in all the progeny (*Figure 4B*), probably because only a small fraction of the wild-type genomes underwent homologous exchange. However, because a low level of the recombinant genome is sufficient to rescue the temperature-sensitive phenotype of $mt:ND2^{del1}$ + $mt:CoI^{T300I}$, this low level of recombinant genome

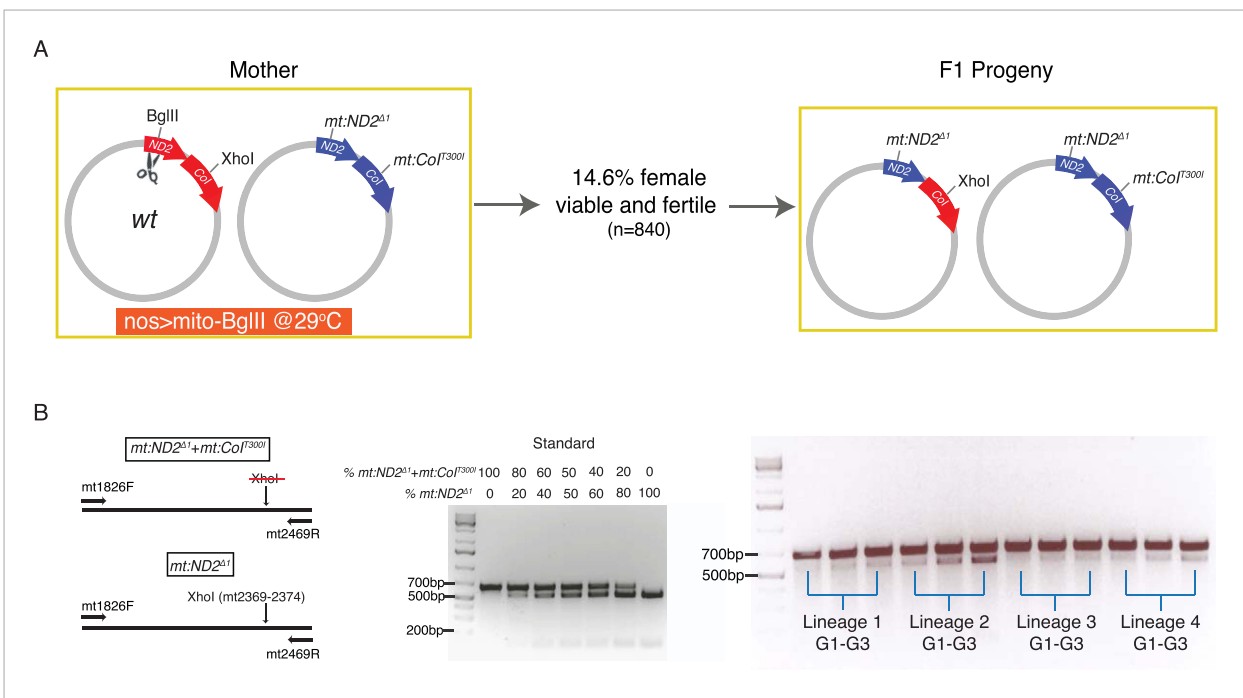

**Figure 4.** Homology-dependent conversion of the wild-type BglII site into the sequence of the $mt:ND2^{del1}$ allele following cutting of the BglII site. (**A**) Expression of mito-BglII in the germline of flies heteroplasmic for wild-type and $mt:ND2^{del1}$ + $mt:CoI^{T300I}$ genomes at 29°C led to isolation of progeny with repaired wild-type genome at the BglII site and converted it to $mt:ND2^{del1}$. (**B**) The starting abundance for the newly generated $mt:ND2^{del1}$ was low, but it increased over generations as the genome possessed a selective advantage at 29°C. The abundance of $mt:ND2^{del1}$ was measured by PCR amplifying a mtDNA region (mt1826–mt2496) using mtDNA from 40 adults as template followed by restriction digestion using XhoI in four heteroplasmic lineages from generation 1 to 3.

supported production of fertile heteroplasmic flies. Since the more functional *mt:ND2^{del1}* genome has a selective advantage at 29°C, its abundance increased over subsequent generations (*Figure 4B*), and in some lineages, the *mt:ND2^{del1}* + *mt:CoI^{T300I}* genome was completely eliminated after 18 generations (*Ma et al., 2014*).

Again, we isolated recombinant genomes at a relatively high frequency, which leads us to propose that a single DSB is sufficient to promote recombination (see 'Discussion'). However, we note that without additional markers, we could do little to characterize the nature of the exchange events.

## Exchange between the mitochondrial genomes of two different species

In order to study whether DSB-induced exchanges involve restricted local repair events, or exchange of longer stretches of DNA, we needed more markers. To achieve this, we examined recombination between the diverged mitochondrial genomes of *Drosophila melanogaster* and *Drosophila yakuba*. We made a heteroplasmic *D. melanogaster* line in which the *D. yakuba* mtDNA is sensitive to BglII and the *D. melanogaster* genome is sensitive to XhoI (see 'Materials and methods') (*Figure 5A*). Within the coding sequences, the *D. yakuba* genome shared about 93% sequence identity with *mt:ND2^{del1}*, while the non-coding AT-rich sequence was highly diverged and reduced to about 1 kb in length. We introduced DSBs into both genomes by germline expression of mito-BglII and mito-XhoI and found that about 10% of the females were fertile. Progeny homoplasmic for recombinant genomes were recovered. All propagating recombinants contained the AT-rich region of the *D. melanogaster* genome (assessed by Southern analysis, *Figure 5—figure supplement 1*), which we expected because *D. melanogaster* genomes outcompete *D. yakuba* genomes very quickly when co-resident (H Ma and PH O'Farrell, unpublished). We characterized the mitochondrial genomes of three progeny lines by PCR and standard sequencing. All the recombinant genomes had one crossover point very close to but upstream of the XhoI site of the *mt:ND2^{del1}* genome (*Figure 5B*). A second crossover is considerably downstream: in two of the recombinants, the other crossover occurred between markers at mt3124 and 3184, and between markers at mt3379 and 3445. In the third case, the other crossover is further downstream and we obtained mixed sequencing signals for a region between mt5524 and mt6291 (*Figure 5C*). Flies were likely to be heteroplasmic for that particular region as if multiple recombinants are carried in this line. Based on these recombinants, we concluded that homology-based repair of DSBs promoted exchange of a substantial uninterrupted stretch of sequence. We also note that there were no discontinuities in the mapped region of exchange as might occur if repair randomly converted mismatches in heteroduplex in one direction or the other (see below for more discussion).

## Discussion

This study, which provides direct evidence for homologous recombination in *Drosophila* mtDNA, opens up the possibility of recombinational mapping of functions on the mitochondrial genome in animals. While the details of recombinational mechanism remain to be worked out, our analysis outlines some of its features. If widespread in animals, the recombinational activity we have observed is likely to have a pervasive influence on the genetics and evolution of metazoan mitochondrial genomes.

### Features of mitochondrial recombination

Multiple modes of homology-dependent exchange are used in different organisms and different situations. Though differing in their details, these different modes often involve DNA breaks of one or both strands, resection of ends, heteroduplex formation, local replication, and resolution. Some features of the processes contributing to mtDNA recombination can be inferred from the sequence of three *D. melanogaster*/*D. yakuba* recombinants that we isolated following introduction of DSBs.

In all cases, one crossover point was very close to the cleavage site of XhoI, indicating stimulation of exchange by DSBs. Some resection (8–44 bp) must have occurred as the sites of exchange are located two or three SNPs upstream of the cleavage site (*Figure 5B*). Surprisingly, the actual exchange events occur in sequences with frequent interruptions of homology such that one of the exchanges occurred within a stretch of only 11 bases of homology. In contrast, some of the more thoroughly characterized homology-dependent mechanisms such as those catalyzed by RecB-C and RecF require significantly greater homology (23–27 bp and 44–90 bp, respectively) (*Shen and Huang, 1986*).

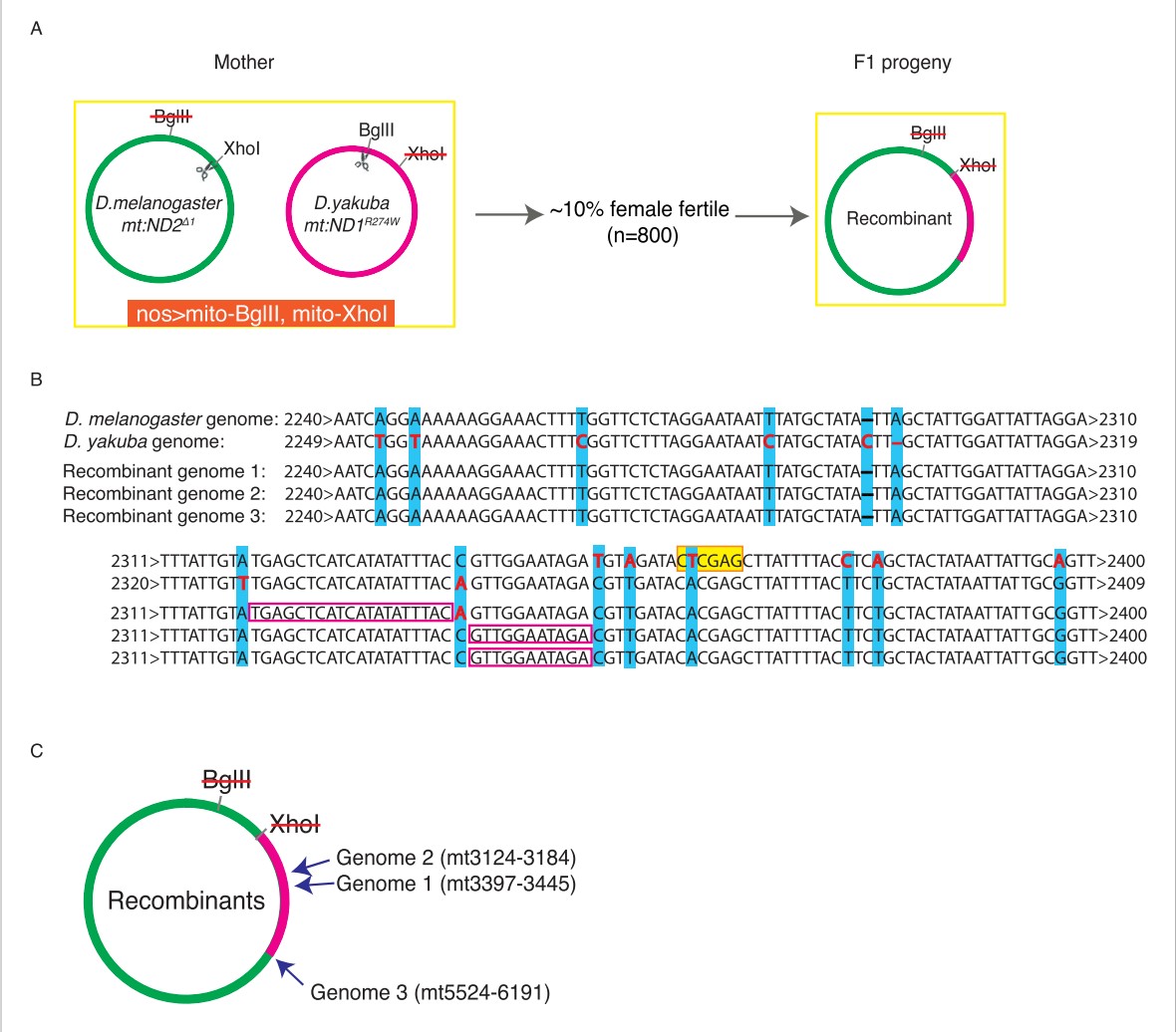

**Figure 5**. Exchange between two mitochondrial genomes of different species that share 93% sequence homology. (**A**) A heteroplasmic line containing both a *Drosophila yakuba* (*mt:ND1^R274W^*) genome and a *Drosophila melanogaster* (*mt:ND2^del1^*) genome was made. 90% of heteroplasmic females induced to express both mito-BglII and mito-XhoI enzymes in the germline were sterile, but the escaper progeny from the fertile females contained newly generated recombinant genotypes. (**B**) Sequences of the parental and three recombinant genomes reveal that one recombination junction is close to the XhoI site of the *D. melanogaster* genome. The XhoI site (yellow highlight) present in the *D. melanogaster* parent sequence (top line) is absent in all the other sequences. Pink rectangles indicate the interval in which recombination occurred. (**C**) A schematic illustration of where the second crossover was for the recombinant genomes. Arrows indicate the approximate position and the numbers, which are based on the *D. melanogaster* sequence, given the positions of bounding polymorphisms.

The following figure supplement is available for figure 5:

**Figure supplement 1**. Southern Blot analysis of the long EcoRI segment covering the AT-rich region of two parental genomes (*D. yakuba* *mt:ND1^R274W^* and *D. melanogaster* *mt:ND2^del1^*) and three recombinant genomes.

This discordance favors other mechanisms such as break-induced recombination (BIR), a template switching mechanism of recombination, which can use very short regions of homology in an exchange process referred to as microhomology-mediated BIR (*Hastings et al., 2009*; *Anand et al., 2014*), to be the underlying mechanism.

The second site of exchange is far from either cleavage site, but seems to occur in regions with high homology. It is not clear whether both exchanges occur at the same time: since the genomes are circular, some types of exchange will produce a dimer circle that might resolve by an intramolecular recombination event at a later time. Our finding that one of our recombinant lines carried more than

one recombinant genome with a common point of exchange near the XhoI cleavage site but different downstream points of exchange is consistent with this possibility. If separated in time, propagation of the unresolved product of the first recombination could give a population of molecules and independent resolution events could give a mixed population. While these analyses give us only a limited perspective on the mechanism of recombination, it is notable that the abundant SNPs suggest that each of the characterized events involved exchange of a substantial segment without interruptions (at least 1 kb), rather than restricted local repair.

When no DSBs were introduced by restriction enzymes in lines heteroplasmic for *ATP6[1]* and *mt:ND2^{del1}* + *mt:CoI^{T300I}*, all of the recombinants again involved the exchange of a substantial and continuous stretch of sequence (at least 5 kb), indicating either the exchange of extended region of duplex or priming of an extended stretch of replication template by a second genome. We also noted that two of the three recombinants had a crossover site very close to where the *Drosophila* mTERF (DmTTF) binds (mt6314–mt6341 and mt11698–mt11725, respectively [*Roberti et al., 2003*]). Although a relatively weak association, we call attention to it because mtDNA replication pauses at the DmTTF binding sites (*Jõers et al., 2013*), and such pauses might destabilize forks and promote exchange.

## The purpose of mitochondrial recombination

Homologous recombination plays major roles in repairing DSBs and in genetic exchange. Below, we discuss the significance of these two roles for maintaining mitochondrial genomes in natural populations.

Our finding that restriction cutting gives frequent recombinant products demonstrates that this process is active in the repair of these DSBs in mtDNA. While genetically distinct genomes in heteroplasmic flies provide the means to detect homologous exchange, sibling molecules ought to have a greater opportunity to guide homology-dependent repair as they will occur, at least transiently, in the same mitochondrion or nucleoid. In homoplasmic flies, accurate repair using sibling molecules will leave no trace. Only misaligned intra-molecular recombination between repeats would give a clear signature. Occurrence of recombination junctions in regions of low homology in this study supports suggestions that non-allelic recombination can contribute to formation of deletions and duplications in the mitochondrial genomes—a frequent cause of mitochondrial defects during aging. Such a recombination could also explain several observations found in natural population (*Rand and Harrison, 1989*; *La Roche et al., 1990*; *Lunt and Hyman, 1997*; *Ludwig et al., 2000*). For instance, mtDNA of European rabbit (*Oryctolagus councils*) has repeated 153-bp motifs in the vicinity of the replication origin of H strand and very individual carries genomes with different numbers of these repeats (*Casane et al., 1994*). In *Drosophila*, mtDNA varies in size from 16 to 20 kb (*Solignac et al., 1986*; *Townsend and Rand, 2004*; *Rand, 2011*), mainly due to the variation in the length of the noncoding region, which contains two types of tandemly repeated elements (*Lewis et al., 1994*). Rand has shown that spontaneous changes in length in populations as if repeats can come and go at a relatively rapid pace (*Rand, 2011*). While accurate repair using sibling molecules is likely to be the main purpose of mitochondrial recombination, more rare non-allelic events can have a major impact on the production of variants, and exchanges between distinctly marked genomes, as reported here, provide a means to characterize the process.

Recombination is detected by and well known for its ability to promote genetic exchange. However, animal mtDNA is restricted in the types of exchange that can occur because uniparental inheritance prevents intermingling of genomes. The mitochondrial genome of each female is sheltered from encounters with foreign genomes and will be passed on in an isolated lineage or clone. Genetic exchange between distantly related genomes might occur as a result of occasional paternal transmission that introduces a foreign genome (e.g., *Kraytsberg, 2004*), but the significance of such violations of uniparental inheritance are matters of an ongoing debate. For the most part, the integrity of mitochondrial haplotypes suggests that the contribution of exchange among distantly related genomes is relatively small (*Elson et al., 2001*).

The limitations on genetic exchange imposed by uniparental inheritance, which are similar to those in asexually propagating organisms, do not mean that there is no meaningful genetic exchange (*Hurst and Peck, 1996*). It has become increasingly apparent that organisms can carry more than one mitochondrial genotype, that is, they are naturally heteroplasmic because of new mutations,

stably transmitting heteroplamic combinations, or a breakdown of uniparental inheritance (*Solignac et al., 1983*; *Ladoukakis et al., 2011*; *Tsang and Lemire, 2011*; *Payne et al., 2012*; *Ma et al., 2014*; *Ye et al., 2014*). In this sense, the situation we created by introducing the temperature-sensitive genome into the *ATP6[1]* line, might be a model for things that can happen naturally. Emergence of a new genome with strong selfish drive, but with a defect in function, or introduction of such genome from another lineage will create a natural situation analogous to our experiment. The outcome of our experimental manipulation shows that this situation has generally detrimental consequences. Indeed, the temperature-sensitive mutation together with the drive advantage created a sort of 'population time bomb': the lineage remained healthy for multiple generations allowing the population to expand greatly, and then it collapsed upon elimination of the functional mt genome. Even rare recombination can uncouple a positively selected drive mutation from detrimental mutations, and as in our experiment, selection can then restore health. Thus, even occasional genetic exchange would prevent rogue genomes from wiping out lineages.

In conclusion, we show that recombination among mitochondrial genomes occurs in *Drosophila* and that this recombination can be used to manipulate genomes for functional mapping. We suggest that recombination will influence evolution of the mitochondrial genome in animals and impact the genetic behavior of mitochondrial disease mutations.

## Materials and methods

### Experimental procedures

#### Fly stocks

The homoplasmic stocks used in this study include the following mutant alleles: $mt:ND2^{del1}$, $mt:CoI^{R301Q}$, $mt:CoI^{T300I}$ and $mt:ND2^{del1} + mt:CoI^{T300I}$. $w^{1118}$ was used to provide wild-type mtDNA. Flies homoplasmic for *ATP6[1]* mtDNA was kindly provided by Michael Palladino (University of Pittsburgh, U.S.). *D. yakuba* flies were obtained from *Drosophila* species stock center, San Diego. Other strains used included *UAS-mito-BglII*, *UAS-mito-XhoI*, *UAS-mito-PstI*, *nos-Gal4*, and *ey-Gal4*. The stocks were cultured at 18–25°C on standard fly medium.

#### Generation of heteroplasmic flies

Poleplasm transplantation was used to generate heteroplasmic flies, and the method was described in *Ma et al. (2014)*. Basically, a portion of the poleplasm was sucked out from donor embryos and transferred into the posterior end of the recipient embryos. The injected recipient embryos were kept in a humidified chamber at 22°C, and hatched larvae were transferred to vials with yeast paste in the next 2 days and incubated at 22°C until eclosion. Lines were established from the females obtained from the injected embryos, which were systematically mated to males with the recipient mtDNA genotype to ensure that the only source of newly introduced mitochondrial genomes was the injected material (i.e., that it did not arise from the purported possibility of exceptional paternal transmission). For each of these females, 10–30 F1 females were isolated to establish sublines. When not specified, each generation was derived from at least 50 individuals belonging to the previous generation.

The procedures to make the heteroplasmic line with both *D. melanogaster* ($mt:ND2^{del1}$) and *D. yakuba* ($mt:ND1^{R274W}$) genomes involve the following four steps. Initially, in order to make a *D. melanogaster* line with the wild-type *D. yakuba* mitochondrial genome, cytoplasm from *D. yakuba* embryos was transplanted into the temperature lethal mutant ($mt:ND2^{del1} + mt:CoI^{T300I}$) embryos, and eclosed adults were kept at 29°C to select for flies with the *D. yakuba* genome. By doing this, two independent lines were established and both stably transmitted *D. yakuba* mitochondrial genome (3–4%) from generation to generation at 29°C. Secondly, a mitochondrially targeted restriction enzyme, mito-PstI, was expressed in the germline mitochondria of the two heteroplasmic lines to eliminate the $mt:ND2^{del1} + mt:CoI^{T300I}$ genome, as PstI site was only present in the *D. melanogaster* genome. Through this, several lines with only wild-type *D. yakuba* mtDNA were established. Surprisingly, *D. melanogaster* flies homoplasmic for *D. yakuba* mtDNA were as healthy as wild-type flies at both 22°C and 29°C. Similar to the *D. melanogaster* genome, wild-type *D. yakuba* mtDNA has one BglII site and one XhoI site. The BglII site is located in the same place as in *D. melanogaster*'s genome (mt:ND2), whereas the XhoI site, unlike the *D. melanogaster* genome (which is in the mt:CoI coding region), is located in the mt:ND1 coding region further downstream. Thirdly, expression of mito-XhoI enzyme in the germline of flies with the wild-type *D. yakuba* genome was used to select

a mutant derivative with an allele $mt:ND1^{R274W}$ that removed the XhoI site. This led to a line that is sensitive to BglII cut but not to XhoI. Finally, this line was then used as a recipient for making the $mt:ND2^{del1}/mt:ND1^{R274W}$ heteroplasmic line.

## DNA isolation

Genomic DNA was extracted from adults as described in *Ma et al. (2014)*. In brief, flies were mechanically homogenized with a plastic pestle in homogenization buffer (100 mM Tris-HCl [pH 8.8], 0.5 mM EDTA, 1.0% SDS). The homogenate was incubated at 65°C for 30 min, followed by addition of potassium acetate (to 1 M) and incubation on ice (30 min) to precipitate protein and SDS. Subsequently, the homogenate was centrifuged at 20,000×*g* for 10 min at 4°C. DNA was recovered from the supernatant by adding 0.5 vol of isopropanol and centrifuging at 20,000×*g* for 5 min at room temperature. The resultant pellet was washed with 70% ethanol and suspended in 100 μl of ddH$_2$O per fly. mtDNA genotype frequencies were measured in individual founding females and their further generations via qPCR. When populations were analyzed, we extracted DNA from groups of 40 individuals.

## mtDNA isolation

Unfertilized eggs were the best source for isolating pure mtDNA without much contamination from genomic DNA: each egg contains 10 ± 2 million copies of mtDNA, which provides about 0.2 ng mtDNA but very little nuclear DNA. In order to isolate a large number of unfertilized eggs, virgins (XX) collected from flies with a certain mtDNA genotype were crossed to males with compound XY (C[1:Y]). Because the progeny of such a cross were male sterile, bulk mating of the progeny gave an abundant source of unfertilized eggs. The unfertilized eggs were then collected and dechorionated with 50% bleach and washed before being homogenized in STE buffer (100 mM NaCl, 50 mM Tris-HCl, pH = 8.5, 10 mM EDTA) using Kimble-Chase Kontes 2-ml glass Dounce tissue grinder. The homogenate was supplemented with 1% SDS and proteinase K and incubated at 55°C for 1 hr. The sample was further treated with RNase A at 37°C for 30 min before phenol–chloroform extraction and ethanol precipitation.

## Southern blotting

Southern blotting was used to detect the recombinant mitochondrial genome. It was performed as described in *Ma et al. (2014)*. Basically, digested DNA was separated on a 0.8% agarose gel by electrophoresis and transferred to Hybond N+ membrane by the capillary method. DNA transferred to the membrane was fixed by UV cross-linking (Stratalinker 120 mJ). The blot was hybridized with PCR-generated probes (mt1577–mt2365) that were labeled with DIG-11-dUTP using 0.35 mM DIG-11-dUTP, 1.65 mM dTTP, and Bioline Velocity Taq DNA polymerase, following the manufacturer's instructions (Roche, Germany). Prehybridization and hybridization were carried out at 41°C overnight in DIG Easy Hyb buffer solution (Roche). The membrane was washed two times with 2 × SSC + 0.1% SDS at room temperature for 5 min and twice for 15 min with 0.1 × SSC + 0.1% SDS at 65°C. Hybridized membrane was visualized with NBT/BCIP following the manufacturer's instructions (Roche). mtDNA primers used to generate the DIG-labeled probe are listed in *Supplementary file 2*.

## Sequencing mtDNA via PacBio SMRT technique

Due to a long repetitive sequence in the AT-rich non-coding region of *Drosophila* mtDNA, PacBio SMRT was used to sequence the whole mitochondrial genome for three genotypes. SMRT sequencing can generate extraordinary long reads (>30,000 bp) with extremely high consensus accuracy (>99.999%). Thus, the whole AT rich region could be covered in a single read without the trouble of re-assembly. Purified mtDNA (isolated as described above) was linearized by restriction cutting at the PstI site and a ~20-kb band was isolated and gel purified after electrophoresis. About 500 ng of gel purified linear mtDNA sample was used for library preparation using a modified 10 kb Template Preparations Protocol (PacBio, University of Washington, Seattle). Basically, the blunt hairpin SMRTBell adaptors were ligated to the repaired ends of the double-strand DNA fragments. Failed ligation products were removed by adding ExoIII and ExoVII exonucleases. The attached templates were further purified with 0.5× and 0.45× bead-washes for sequencing. We conducted the sequencing reactions on a PacBio RSII system using one SMRT cell for each genome. All libraries were sequenced using P4/C2 chemistry. mtDNA samples from $mt:ND2^{del1} + mt:CoI^{T300I}$, and the recombinants were

run with 120 min movie time and *ATP6[1]* sample was run with 180 min. Raw reads were analyzed following either HGAP (for de novo assembly) and BLASR protocol (for resequencing) in SMRT Portal 2.2. The coverage for each samples ranged from 1000× to 35,000×.

When only the coding region of some genomes were sequenced, two long-range PCR reactions using Expand Long Template PCR system (Roche) was performed: mt186–mt7502, mt6905–mt14797 with the following program: 1 cycle of 93°C for 3 min, 30 cycles of 93°C 15 s, 50°C 30 s, 60°C 8 min, and 1 cycle of 60°C for 10 min. Primers were designed all around the coding region (*Supplementary file 2*) for sequencing by QuintaraBio (Albany, CA).

## Acknowledgements

This research was supported by NIH (GM086854) funding to PHO'F. HM was supported by the Long-term postdoc fellowship from Human Frontiers Science Program. We thank Hong Xu for generously sharing unpublished information and reagents, and Michael J Palladino at University of Pittsburgh for kindly providing us flies with *ATP6[1]* genome. We also are grateful for support from the PacBio team, particularly Nicole Rapicavoli (Melon Park, CA), Maika Malig (University of Washington, Settle), and Roberto Lleras.

## Additional information

### Funding

| Funder | Grant reference | Author |
| --- | --- | --- |
| NIH Office of the Director | GM086854 | Patrick H O'Farrell |
| Human Frontier Science Program | LT000138/2010-L | Hansong Ma |

The funders had no role in study design, data collection and interpretation, or the decision to submit the work for publication.

### Author contributions

HM, Conception and design, Acquisition of data, Analysis and interpretation of data, Drafting or revising the article; PHO'F, Conception and design, Analysis and interpretation of data, Drafting or revising the article

### Author ORCIDs

Patrick H O'Farrell, http://orcid.org/0000-0003-0011-2734

## Additional files

### Supplementary files

• Supplementary file 1. HR-based repair was also able to rescue in somatic tissues like eye. High-level eye-antennal disc expression of both mito-BglII and mito-XhoI using ey-Gal4 at 29°C resulted in pupal lethality (headless pupa) in wild-type flies or in flies carrying mt genomes resistant to only one of the two enzymes. A few (about 1%) escapers eclosed and the majority of eclosed flies had no eye or small eye phenotype (rows 1 and 2). Having both genomes increased the eclosion rate to 10% (row 3) and most of the eclosed flies had normal-sized eyes (not shown).

• Supplementary file 2. A list of primers used in this study.

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
