## [Decision Letter]

Thank you for sending your work entitled “Selections that isolate recombinant mitochondrial genomes in animals” for consideration at *eLife*. Your article has been favourably evaluated by Detlef Weigel (Senior Editor) and three reviewers, one of whom is a member of our Board of Reviewing Editors. One of the three reviewers, David Rand, has agreed to share his identity.

The manuscript describes a *Drosophila* animal model that can generate genetic exchange between mitochondrial genomes, which the reviewers felt was fully supported by evidence. The reviewers also unanimously agreed that the development of a system that overcomes the existing mtDNA haplotype “barrier” represents a significant advance in the field. However, the mechanistic basis of the genetic exchange observed was not sufficiently elucidated. In addition, the reviewers noted that the genetic exchange events observed in the studies were rare and the frequency of such events was only observed under conditions of strong selection, raising the concern that the system does not inform the mechanism of native events. On a related note, the reviewers also noted that that most of the recombination observations reported in the manuscript have been previously published, which was not adequately acknowledged. Given these points, it was decided that the advance lies in the development of a workable system for studying mtDNA genetic exchange, which could be used, for example, to map regions that are causative of disease in an animal by the community. As such, the consensus was to ask the authors to revise their study as a “Tools and Resources” manuscript.

Comments on mechanistic aspects of the work:

Data clearly support the appearance of mtDNA haplotypes containing tracts of sequence from the two parental mtDNA molecules following selection. However, whether these haplotypes are products of homologous recombination is not sufficiently clear. For example, the authors have not definitively excluded gene conversion as a possible explanation for their results. The length of the putative crossover tract is not itself sufficient to distinguish between gene conversion and true crossover events. Accordingly, the authors should revise their title to refer to these events more generally as genetic exchange.

There are well-established statistical tests for gene conversion that should be applied, especially considering that in some cases the putative recombinant haplotypes have already been sequenced.

The authors have not established that parental haplotypes are in close proximity required for recombination (such as by imaging heteroplasmic *Drosophila* nucleoids labelled by FISH), despite the ability to do so as demonstrated in previous manuscripts from this group.

The authors have not provided any evidence that mtDNA of *Drosophila* is capable of forming true Holliday junctions, and failed to discuss relevant literature on the structure of mtDNA replication intermediates.

Crossover points in putative recombinants are often in the vicinity of the heavy or light strand replication origins or the mTERF replication pausing site. How did the authors exclude the possibility that novel haplotypes may arise via strand-switching of stalled forks? Can other restriction sites be used to induce DBSs that are further from these sites?

The authors' assertion that the use of restriction-targeted enzymes actually informs us about the mechanism of the unassisted recombination events is not necessarily true. Introduction of the strand breaks may also result in exchange, but this correlation does not prove that the same mechanisms actually apply under the restriction digestion-free situations. As such, the authors should revise this claim. In addition, the authors also imply that recombination is somewhat common, and that the selection provides a means to “amplify” and detect these events. Have the authors considered that the negative physiological consequences of mitochondria under these selection regimes may be required to generate the recombination positive molecules?

Comments on the novelty and presentation of the work:

The reviewers felt that while data clearly demonstrate that selection is critical for exchange in the fly, most of the observations on exchange have been previously published. Additionally, these observations have been made before, in hybrid systems, but used amplification-based methods which may have induced artifacts (e.g. Ujvari, B et al., 2007, Biol. Lett., 3, 189-192; [18], Genetics, 172, 1745-1749), but were robustly demonstrated in the bi-uniparental mitochondrial systems (e.g.. [27], Mol. Biol. Evol., 28: 1847-1859), at a very low rate. On a related note, the reviewers also request that the Discussion be revised, specifically to eliminate overstatements of significance. The reviewers noted that a significant fraction of the Discussion was dedicated to a potential explanatory mechanism of exchange that may not actually be occurring in a native system as the double-strand break model engineered with restriction enzymes is highly artificial, and may not actually be analogous to the natural mechanism.

Minor comments:

The discussion of the results in the beginning of the subsection “Screening for recombination without selection” (Results) is confusing. In their Ma et al., Nature Genetics 2014 paper, the authors write: “Lines in which the temperature-sensitive genome was paired with the *mt:ND2*^*del1*^ genome also showed an early decrease in the abundance of the temperature-sensitive genome. However, the decrease in abundance did not continue to 0% but asymptotically approached ∼8%.” Does this observation mean that there actually is selection at the mtDNA level under the “selection free” scenario? If true, this implies that these events are only observable under scenarios of near-lethal conditions, not simply under selection.

The paragraph “If it were to occur […] degenerative changes of this genome” is somewhat overstated. Given the current estimated rates of mutation, any two mtDNAs would differ by only a single SNP.

The authors overstate the results of this work (for example, in the sentences “The finding means that recombination […] mitochondrial disease” and “We suggest that recombination […] mitochondrial disease mutations”. The events observed occur only under extreme selection regimes, and between highly divergent molecules.

In the subsection “The purpose of homologous recombination” of the Discussion, the authors state: “… rabbit (*Oryctolagus cuniculus)* mtDNA have repeated 153 bp motifs in the vicinity of the replication origin of H strand…”. Would slip-strand mispairing not fully explain this pattern? This activity is quite common in animal mtDNA.

In the same subsection, the authors state that: “recombination among sibling molecules may be prevalent”. Why claim recombination is prevalent? The data in the Results show the opposite.

In your manuscript, you suggest that: “the main role of recombination in mitochondria is homology dependent DNA repair, which may be especially important in light of high level of DNA damage inflicted on the mtDNA by its oxidative environment”. There is growing skepticism regarding the amount of ROS damage that mtDNA experiences (example: PLoS Genet. 2014 Feb; 10(2): e1003974).

Please clarify the following note (in the subsection “Recombination upon introduction of a single DSB”): “genome was completely eliminated after 18 generations (not shown)”. Why “data not shown”? Why not show data?

Figure 1: The size of the band in Figure 1 looks bigger than 21.2kb. The literature (and this paper) indicate that *D. melanogaster* mtDNA is ∼19kb. Clarify this discrepancy. *D. melanogaster* mtDNA size does vary, but >21.2 kb is an unusually large size.

Figure 2: Where are the other digestion products on the gel. There should be a ∼3kb band produced by this digest. It looks like the probe does not overlap this smaller band. It would be reassuring that all products from the digest are demonstrated on the Southern blot. Can these blots be re-probed with another probe? A PCR product for the targeted region to the right of the Xhol site would confirm this.

Discussion: Can some estimates be made about the frequency of exchange in the heteroplasmic cultures? A big issue in this new finding is just how frequent this is in nature. The Discussion does a good job of clarifying that recombination will not happen without paternal leakage, mtDNAs being of different haplotypes, and being in the same organelle or ‘nucleoid’ of mtDNAs. There are two issues of interest and impact here. First, population geneticists might want to know: how much recombination is enough to purge deleterious mutation accumulation? Second, molecular geneticists will want to know how often these occur for disease issues or for constructing novel genotypes for experimental work. This frequency estimate is a rough guess, but that would increase the impact.

---

## [Author Response]

*The manuscript describes a* Drosophila *animal model that can generate genetic exchange between mitochondrial genomes, which the reviewers felt was fully supported by evidence. The reviewers also unanimously agreed that the development of a system that overcomes the existing mtDNA haplotype “barrier” represents a significant advance in the field. However, the mechanistic basis of the genetic exchange observed was not sufficiently elucidated. In addition, the reviewers noted that the genetic exchange events observed in the studies were rare and the frequency of such events was only observed under conditions of strong selection, raising the concern that the system does not inform the mechanism of native events. On a related note, the reviewers also noted that that most of the recombination observations reported in the manuscript have been previously published (see below for details), which was not adequately acknowledged. Given these points, it was decided that the advance lies in the development of a workable system for studying mtDNA genetic exchange, which could be used, for example, to map regions that are causative of disease in an animal by the community. As such, the consensus was to ask the authors to revise their study as a “Tools and Resources” manuscript*.

This overview suggests that “the mechanistic basis of the genetic exchange observed was not sufficiently elucidated”. We acknowledge that there is much to learn about the mechanism of recombination. However, we report unprecedented advances documenting and characterizing recombination in animal mitochondria and fully document our findings.

The overview also notes that the recombinants were rare and observed “under conditions of strong selection”. Recombination has been studied for decades by selection for recombinant genotypes and in many circumstances the events are rare (e.g. somatic recombination leading to loss of heterozygosity and cancer progression is fortunately particularly rare). For reasons that are not clear to us, our application of a genetic approach raises a heretofore never mentioned concern “that the system does not inform the mechanism of native events”. Is this an indictment of the field of genetics or just our work? We fail to understand the logic. Neither rarity nor detection by genetic selection should disqualify recombination from being a natural event. Recombination is often a rare event, but nonetheless has profound consequences and is thought to be an essential to avoid genomic deterioration by “Muller’s ratchet” (35). Furthermore, selection is usually thought to act on pre-existing events, which should qualify as natural, after all Darwin called it natural selection. Nonetheless, the possibility that stress would induce recombination is real, but it adds to, rather than diminishes, the biological impact of the process. Recombination among fully functional genomes can generate non-allelic events that cause mutageneic rearrangements, so it might be something to avoid in the absence of problems. In contrast, recombination can restore function to genomes with distinct defects and so might be productively induced under circumstances of insufficient function or DNA damage. We consider this further below in responding to a specific comment in the review.

The reviewers “noted that most of the recombination observations reported in the manuscript have been previously published, which was not adequately acknowledged”. Firstly, the reviewers mention papers that we did reference. While there are many additional and unreferenced papers in the literature that discuss recombination in animal mitochondria, we believe that we appropriately represented the controversial status of the literature. Secondly, the review does not point out any observation that anticipated a key finding of this manuscript. The only reported observation that has a parallel in the literature is one that we presented for the express purpose of showing a parallel. We showed that heteroplasmic strains can faithfully propagate two genomes for multiple generations. Like a similar result in mice, our analysis failed to detect recombinants in flies without selection. It was presented to suggest potential parallels between systems and to highlight the importance of the special tools that we developed to take the analysis of recombination to a new and unprecedented level. The central approaches, the outcomes and the characterizations that are presented in this paper are unique. We discuss the specific papers below, but note here that they show little consensus about the existence of recombination in animal mitochondria and have little to say about its characteristics.

*Comments on mechanistic aspects of the work*:

*Data clearly support the appearance of mtDNA haplotypes containing tracts of sequence from the two parental mtDNA molecules following selection. However, whether these haplotypes are products of homologous recombination is not sufficiently clear. For example, the authors have not definitively excluded gene conversion as a possible explanation for their results. The length of the putative crossover tract is not itself sufficient to distinguish between gene conversion and true crossover events. Accordingly, the authors should revise their title to refer to these events more generally as genetic exchange*.

Recombination is a genetic term that refers to the appearance of non-parental combinations of input alleles in progeny organisms or cells. This is what is reported in the manuscript. Like recombination, gene conversion has a genetic definition: when the products of meiosis reveal allele frequencies that are inconsistent with Mendel’s rules, one allele is said to be converted to another. Gene conversion represents a type of recombination, i.e. nonparental combinations of alleles are found in progeny. For example, Chen et al. (Nature, 2007) state: ‘In eukaryotes, gene conversion constitutes the main form of homologous recombination that is initiated by DNA double-strand breaks.’ Additionally, according to Molecular Biology of the Cell. 4th edition (Bruce Alberts et al): ‘In the process of gene conversion, DNA sequence information is transferred from one DNA helix that remains unchanged (a donor sequence) to another DNA helix whose sequence is altered (an acceptor sequence). There are several different ways this might happen, all of which involve the following two processes: (1) a homologous recombination event that juxtaposes two homologous DNA double helices, and (2) a limited amount of localized DNA synthesis, which is necessary to create an extra copy of one allele.’ The fact that recombination is a term that encompasses gene conversion is nicely highlighted by the fact that Robin Holliday’s presentation of his model for recombination (1964 Genet. Res., 5, pp. 282–304) was entitled “A mechanism for gene conversion in fungi”. We thus do not agree with the reviewers’ concern about revising the title.

We are not quite sure what is meant by the comment that “whether these haplotypes are products of homologous recombination is not sufficiently clear”. Again, recombination is defined by recovery of non-parental recombinant progeny and we demonstrate this at a phenotypic and genetic level. Homologous recombination refers to events in which the products are joined in regions of homology. The precise allelic junctions in the multiple recombinants isolated show that these recombinants are the result of homology directed exchange. We believe that this is clear. If the issue is that we have not defined the detailed enzymology and mechanism, we agree, but do point out that in those systems where recombination has been studied for decades, mechanisms were worked out over many years by many laboratories, and still recent work is refining our views of mechanism in even the most studied systems. Additionally, we emphasize that recombination is not defined by a unique mechanism. Indeed, it occurs by multiple mechanisms. For example, bacteriophage Lambda recombination, which is one of the pillars of modern genetic engineering by recombination, recombineering (e.g. Pines et al., 2015), does not use the same mechanism as *E. coli* RecA mediated recombination. Lambda relies heavily on a processive exonuclease that produces extended tracks of single stranded DNA that interact, while RecA mediates pairing between duplex molecules. We argue that we have made unprecedented advances in documenting and characterizing animal mitochondrial recombination and acknowledge that the system is in its infancy and there is still more to learn.

The review suggests that an example of the insufficiency of the work is the failure to definitively exclude gene conversion as a source of the recombinants. This focus on gene conversion appears to be based on misunderstanding that gene conversion is a mechanism. As mentioned above, gene conversion is defined by change of one allele into another, and it can occur by multiple mechanisms. For example, translation of Holliday junction results in heteroduplex formation with mismatches at points of sequence discordance: Mismatch repair can rectify the mismatches, most frequently by excision and repair, resulting in the loss of one allele (Holliday, 1964). Alternatively, in double-strand break induced recombination, heteroduplex can also be produced in other ways and similarly be repaired by mismatch repair mechanisms (Kobayashi, 1992). Additionally, a synthesis dependent strand annealing mechanism (McMahill et al., 2007) provides yet another mechanism’s generating heteroduplex. In sum, gene conversion is not defined by mechanism, and there exist multiple mechanisms leading to conversion. Furthermore, the mechanisms do not have a simple one to one relationship with genetic outcomes, with one producing recombination and the other gene conversion. Importantly, all events leading to recombinant genomes are appropriately referred to as recombination. In reviewing our text, we realized that in several places we unintentionally suggested the dichotomy between recombination and gene conversion that we now challenge. We have re-written several paragraphs and changed a section subtitle in the Discussion to avoid this.

There are well-established statistical tests for gene conversion that should be applied, especially considering that in some cases the putative recombinant haplotypes have already been sequenced.

There have been detailed characterizations of gene conversion in particular biological settings, but the measured parameters do not result in a “test for gene conversion” as the values are largely dictated by conversion track length, a parameter that varies greatly with context and specific mechanism underlying the conversion process. For this reason, we do not feel that the suggested analysis would be valid. Nonetheless, it is true that the commonly studied forms of gene conversion have relatively short track lengths and we have considered whether our results are congruent with other cases of gene. The mean tract length for meiotic gene conversation has been estimated by several experimental studies for organisms including yeast, human and *Drosophila* to be in the range of 350-2000bp (Padhukasahasram and Rannala, 2013, Chen et al, 2007). In *Drosophila*, it is estimated to be ∼352bp (Hilliker et al., 1994). The recombinants we isolated between *D. mel* and *D. yak* all involve at least 1kb of continuous exchange. Furthermore, the recombinant we isolated between the temperature sensitive mutant and *ATP6[1]* showing exchange of a continuous 7.5kb fragment, which is well above the mean tract length of nuclear meiotic gene conversion (using the data and model of Hilliker et al., 1994 for meiotic nuclear events, the probability of a gene conversion track of this length would be less than 10^-9^). More recently, by following more lines heteroplasmic for *ATP6[1]* and temperature sensitive genomes, we recovered two more recombinants (this finding and sequence data have been added to the manuscript as part of Figure 2—figure supplement 1), both involved exchange of a long stretch of continuous sequence (5-12kb), again well outside of the range characteristic of most gene conversion systems. However, since different mechanisms of homologous recombination produce products that are conservative exchange events as well as products that convert one allele to another, we suggest that efforts to distinguish these processes without the genetic test that defines them is not terribly important. Instead, the important aspects that we report are the features of the recombination products. We have re-written the Discussion to maintain a more direct connection to the data provided and believe that these data provide meaningful insights into processes involved in generating the recombinants that we describe.

*The authors have not established that parental haplotypes are in close proximity required for recombination (such as by imaging heteroplasmic* Drosophila *nucleoids labelled by FISH), despite the ability to do so as demonstrated in previous manuscripts from this group.*

We do not consider that it is necessary to show that parental haplotypes are in close proximity required for recombination, because we documented that they did recombine. We have never imaged heteroplasmic *Drosophila* nucleoids labelled by FISH in any of our previous manuscripts. We also note that the interaction that gave rise to the recombinants might have been transient, and might have occurred at any time during the life cycle of the flies. Lastly, at this point in time, we believe the imaging criterion that would define the proximity for recombination is not practical and we do not see how its absence calls into question the existence of the recombinants that we have characterized.

*The authors have not provided any evidence that mtDNA of* Drosophila *is capable of forming true Holliday junctions, and failed to discuss relevant literature on the structure of mtDNA replication intermediates*.

These junctions have seldom been visualized outside of their generation in in vitro reactions and the lack of direct demonstration in most systems has not prevented study of recombination. Additionally, Holliday junctions are not used in all forms of recombination and the structure can also form without generating a recombinant. We feel that a demonstration of Holliday junctions is impractical at this stage and largely tangential to the advances made in the paper.

We were reticent to discuss the structure of mtDNA replication intermediates because there is little consensus in *Drosophila* even about major aspects of the mechanism of replication. For instance, a recent paper by Joers and Jacob, 2013, argues that replication is unidirectional (starting within the non-coding region and proceeding in the direction of the rRNA locus) and mainly strand-coupled (as almost all intermediates that were detected by 2D electrophoresis were found to be fully double-stranded). However, their data disagree with the previously proposed strand-displacement model for mtDNA replication, which is based on transmission electron microscope imaging of replicative forms (e.g. Goddard and Wolstenholme, 1978). We remain of the opinion that there is little that would be gained from an extensive discussion of potential interplay with replication at this time, but in response to the next question we have added specific considerations and tried to make more clear that we have not excluded interactions with replication.

Crossover points in putative recombinants are often in the vicinity of the heavy or light strand replication origins or the mTERF replication pausing site. How did the authors exclude the possibility that novel haplotypes may arise via strand-switching of stalled forks? Can other restriction sites be used to induce DBSs that are further from these sites?

Our manuscript does not eliminate particular mechanisms, and we do not exclude strand-switching mechanisms. Indeed, we suggested “break induced replication” or BIR as a possible mechanism. This mechanism posits that breaks induce invasion of homologous duplex, followed by replicative extension of the invading strand, that is strand switching (see Arand et al., 2013). Although, the name emphasizes double-strand breaks and we advanced this particular idea because of data obtained in double restriction enzyme selection, the mechanism also appears to be used to repair stalled forks (Constantino et al., 2014).

The suggestion that we examine possible coincidence of recombination junctions with special sequences such as replication origins is interesting, but without tests of causality it is not clear that much meaning can be attached to it. Nonetheless, we examined this issue. Most of our sites of recombination have not been localized near an origin of replication, but this might be influenced by rapid divergence of sequences around the replication origins so that homology is reduced. None of the most accurately mapped sites of recombination, those between *D. melanogaster* and *D. yakuba*, map to any of the mentioned sites. However, the location of sites of recombination in these cases might be special because sites are restricted by the frequent interruptions of homology in this pairing. One junction of the recombinant between *ATP6[1]* and the temperature sensitive genome was mapped within a 531 bp region (1154-12085) that encompasses a mTERF binding site downstream of ND1. Additionally, one of two other newly isolated recombinants between these genomes was mapped within a 631 bp region (5978-6619) that encompasses the other site of mTERF binding at 6314. While possibly consistent with a role of replication fork stalling near mTERF sites in triggering recombination in two cases, the correlation is weak and several steps of inference are involved in making such a suggestion. We have added a short discussion in the manuscript.

The authors' assertion that the use of restriction-targeted enzymes actually informs us about the mechanism of the unassisted recombination events is not necessarily true. Introduction of the strand breaks may also result in exchange, but this correlation does not prove that the same mechanisms actually apply under the restriction digestion-free situations. As such, the authors should revise this claim. In addition, the authors also imply that recombination is somewhat common, and that the selection provides a means to “amplify” and detect these events. Have the authors considered that the negative physiological consequences of mitochondria under these selection regimes may be required to generate the recombination positive molecules?

The issue that restriction enzyme cutting might induce a distinct process is a reasonable caution and we made changes to in the discussion to accommodate this criticism. Additionally, we have deleted the word natural in the Abstract, since it had inappropriately implied that the restriction cuts were directly stimulating the normal event.

We totally agree that stress might induce recombination, but it is difficult to test. We did attempt to do so as follows. For the *ATP6[1]* and temperature sensitive selection, the five lines that we followed at 29°C were also followed at 22°C (Populations in each line were split into half at generation 1 and kept at either 29 or 22°C for the following generations). At 22°C there is little selection against the temperature sensitive genome and presumably little stress as the flies are healthy. At generation 5 (when the average abundance of the *ATP[1]* genome had dropped to about 10% in most lines), populations growing at 22°C were transferred back to 29°C and kept there for the subsequent generations. We had reasoned that if the recombinants were pre-existing, then one would expect the same outcome whether the selection was imposed from the beginning or only after the decline the *ATP6[1]* genome. Perhaps supporting such a notion, none of the lines that were initially at 22°C produced survivors, suggesting the possibility of induced recombination by negative physiological consequences at high temperatures. However, as we considered events during the selection, we realized the finding might be explained without assuming induced recombination. Because there is huge growth of the population during the early generations, we can only analyze a small fraction of the potential progeny (we cull the population at each generation). Hence, the temperature might also have had an affect on the efficiency of recovery of preexisting recombinants by giving them an early selective advantage so that they were less likely to be culled in the early generations. Since the above experiment does not provide a definitive conclusion, we did not include it in the manuscript. Thus, the idea while perhaps weakly supported by our experiment, remains hypothetical. As emphasized in our comments above, stress-induction of recombination might add greatly to its impact. Consequently, it is a factor we do not want to ignore but will continue to investigate.

*The reviewers felt that while data clearly demonstrate that selection is critical for exchange in the fly, most of the observations on exchange have been previously published. Additionally, these observations have been made before, in hybrid systems, but used amplification-based methods which may have induced artifacts (e.g. Ujvari, B et al., 2007, Biol. Lett., 3, 189-192;*
[18]*, Genetics, 172, 1745-1749), but were robustly demonstrated in the bi-uniparental mitochondrial systems (e.g..*
[27]*, Mol. Biol. Evol., 28: 1847-1859), at a very low rate. On a related note, the reviewers also request that the Discussion be revised, specifically to eliminate overstatements of significance. The reviewers noted that a significant fraction of the Discussion was dedicated to a potential explanatory mechanism of exchange that may not actually be occurring in a native system as the double-strand break model engineered with restriction enzymes is highly artificial, and may not actually be analogous to the natural mechanism*.

The reviewers challenged the novelty of the work because they felt that most of the observations on exchange have been previously published. Except for [18], the work they quote is work that we reference. Here we examine these reports to consider whether they demonstrate that our observations “have been previously published”.

The Larsson group presents its conclusion in the title ‘No recombination of mtDNA after heteroplasmy for 50 generations in the mouse maternal germline’. Clearly this paper did not report genetic selections for organisms carrying characterized recombinant genomes. Indeed, just the opposite, it argued that there is no recombination. As mentioned above, we did report a parallel analysis in *Drosophila* with a similar outcome. We reported this experiment because we suspect that that there is little difference between systems other than the fact that we developed uniquely powerful tools that allowed us to go beyond this limited experiment to do heretofore never reported experiments directly selecting and isolating animals with recombinant mitochondrial genomes.

A. [41], in contrast with S. [6], reports mitochondrial recombination in mice. Sato et al. made heteroplasmic tissue culture cells and heteroplasmic mice and retrieved mitochondrial genomes from the heteroplasmic lines by cloning. Many independent clones were sequenced. In one of three experimental settings, they identified two rare clones having short stretches of sequence with SNIP markers from the complementary genotype. This paper nicely shows rare somatic recombinant molecules that include between 36 bp to 120 bp of transferred sequence. Notably, the sequences transferred are short. There is no demonstration of germline events, no transmission of the recombinants and no enrichment of the recombinant. Furthermore, the rare events they detect have no demonstrated functional consequence. Outside of an evolutionary time scale, we consider this the clearest published experimental test for recombination in animals prior to our work, but none of the reported experiments resemble ours and their results are more limited and different.

H. Fukui et al., 2009 made heteroplasmic tissue culture cells and mice and used PCR reactions to support claims for rare recombinant molecules provoked by expression of ScaI restriction enzyme. Unfortunately, PCR reactions are themselves recombinogenic and both of the above papers show that this gives rise to artifactual results. In addition to uncertain validity, this study only detected genomes with big deletions close to the cut site and all but one of the apparent deletions was interpreted as intramolecular. Instead of full genomes resulting from exchange of sequence between the parental genomes, rearranged (deleted) sequences that were recovered by PCR support this recombination event. There is no evidence of germline transmission, no evidence of functional contribution, and uncertainty about the contribution of multiple restriction sites to the generation of the reported fragments. While this report might overlap in its intent, it does not report observations comparable to those in our paper.

B. [49] reports an analysis of the mitochondrial genome sequence of wild lizards captured at a boundary between two different populations of lizards, which have mitochondrial genomes marked by many sequence differences. A single lizard was found to have a genome with a somewhat intermediate sequence. Analysis of the distribution of polymorphisms in this intermediate genome suggests an evolutionary connection to the two dominant haplotypes, where one segment was highly related to one haplotype and the other segment was related to the other haplotype. This strongly suggests that a recombination event occurred at some time in the evolution of the intermediate haplotype. However, numerous sequence differences indicate that the dominant existing haplotypes were not the immediate parents of the intermediate haplotype. Thus, the study does not identify the parental genomes, but infers that a recombination event at some time in evolution (apparently long ago) was the source of the exceptional genome. Such population studies of wild populations, while of value and supporting the idea that some genetic exchange exists, bare little resemblance to our analysis, and do not report observations that we have made.

The Guo et al. paper has some parallels to B. [49], but we feel is much less definitive. The Guo paper examines the sequences of carp that are used in farmed fish production in China. An interesting set of cross-species crosses generate hybrid fish with useful attributes. The paper reports analysis of the mitochondrial DNA from a number of fish selected from the populations used in the final cross as well as several hybrid fish. One of the hybrid fish has a mitochondrial genome sequence unlike that of the other hybrids or of the presumed parents. An argument is presented that this genome is recombinant between the two parental types. However, the relationship of the sequence to presumed parents is very inexact with many polymorphisms distinguishing the “recombinant” genome from either parent. As result, it is clear that the genome was not produced as an immediate product of recombination and given the lack of knowledge of the lineage of the fish analyzed and cross species cross used to produce the populations examined, it seems likely that this diverged genome was introduced by introgression form another strain or species.

A. [41] reports another analysis of wild caught organisms. In this case, the clams that were examined exhibit an exceptional and interesting type of inheritance pattern of their mitochondrial genomes called doubly uniparental. Here the male mitochondrial genome in the sperm is transmitted to the progeny, but its contribution is kept compartmentalized and separate from that of the mitochondrial genome from the egg. The male mitochondrial genome contributes only to the mitochondria in the male germline. The separation between the female lineage and male lineage is so thorough that they evolve along different paths and have dramatic differences in sequence suggesting isolation for hundreds of thousands to millions of years. Nonetheless, some populations have male genomes with segments having a high similarity to the female genome, a finding that is taken as indication of recombination event in more recent evolutionary history. This is a very interesting biological system and the study supports rare recombination in this unusual setting. However, it implicates a recombination event in the distant past, albeit more recent than the original separation between the paternal and maternal lineages. As in the above cases, we disagree that this paper can be presented as evidence that “most of the observations on exchange have been previously published”.

The cited work do not report experiments analogous to ours, or show findings analogous to ours. If the reviewers’ point is that there was prior evidence for recombination in mammals, we do not dispute this. Indeed, we described the literature as controversial, so of course it contains papers arguing for both interpretations. Finally, we would like to declare that, to our knowledge, there has been no previous case in animals in which recombinant mitochondrial genomes have been selected in animals and progeny recovered with a dominant recombinant genome.

*Minor comments*:

*The discussion of the results in the beginning of the subsection “Screening for recombination without selection” (Results) is confusing. In their Ma et al., Nature Genetics 2014 paper, the authors write: “Lines in which the temperature-sensitive genome was paired with the* mt:ND2^del1^
*genome also showed an early decrease in the abundance of the temperature-sensitive genome. However, the decrease in abundance did not continue to 0% but asymptotically approached ∼8%.” Does this observation mean that there actually is selection at the mtDNA level under the “selection free” scenario? If true, this implies that these events are only observable under scenarios of near-lethal conditions, not simply under selection*.

This is a very good point, and there is likely some truth in the inference that is made. In the line heteroplasmic for *mt:ND2*^*del1*^ and *mt:CoI*^*T300I*^ genomes, there appears to be is selection against both genomes to achieve the balanced ∼8% ratio. However, this balanced heteroplasmic line behaved, if not better, at least as well as the flies homoplasmic for wild type mitochondrial genome (See supplementary Table 1b in Ma et al., Nature Genetics 2014). Therefore, there is no functional advantage to the organism incurred by generating a wild type recombinant mtDNA under such a condition. Thus, from an organismal perspective the condition appears to be ‘selection free’, but there does seem to be selection acting on the mitochondrial genomes to maintain the ratio (as pointed out). Because there is some level of selection ongoing in this heteroplasmic line, we have changed the title of the first section of the results as the previous wording had implied otherwise.

Why wouldn’t this selection on the mitochondrial genomes act to promote formation of recombinants during this “selection free” situation? One reason is that recombination might not occur at the same stage of the life cycle as the purifying selection, which is limited to oogenesis. Indeed, in order to give functional genomes an advantage over less functional genomes during oogenesis, they must be autonomous and presumably in different mitochondria, while recombination requires that they reside in the same mitochondria. In accord with the idea that different types of selection can occur at different times, we find that replicative drive imparts a selective advantage at stages other than oogenesis (Ma and O’Farrell, submitted). Hence, the likely answer to this vexing issue is that purifying selection and recombination may not happen at the same time.

*The paragraph “If it were to occur […] degenerative changes of this genome” is somewhat overstated. Given the current estimated rates of mutation, any two mtDNAs would differ by only a single SNP*.

Apparently our wording suggested more than we intended. We have changed it.

We do agree with the reviewers that more recent and seemingly more definitive measures have suggested rather modest frequencies of SNPs (e.g. Kennedy et al., 2013) compared to some suggestions in the literature. However, this does not preclude important actions of recombination either somatically or in the germline. There are strong data indicating that deletions of mtDNA increase in incidence with age, that they accumulate to high levels in affected cells, and that the end points of the deletions tend to lie in regions of local homology suggesting that recombination contributes to their formation (e.g. [46]; S. [34]; Bacman, Williams, and Moraes, 2009). Additionally, direct analysis of mtDNA from individual intestinal crypts shows that these rapidly dividing cells accumulate a few SNIPs to relatively high abundance and likely carry many more at lower levels (Taylor et al., 2003).

*The authors overstate the results of this work (for example, in the sentences “The finding means that recombination […] mitochondrial disease” and “We suggest that recombination […] mitochondrial disease mutations”. The events observed occur only under extreme selection regimes, and between highly divergent molecules*.

For the first passage, the concern that we made statement about normal processes from an experiment that involves a perturbation applies, and we have changed it. We have altered the wording to make clear that it is a possibility but not a conclusion from the results.

Regarding the second sentence, it is not a conclusion. It is a suggestion that recombination will be influential as it is throughout biology, and we do not regard this as an overstatement.

*In the subsection “The purpose of homologous recombination” of the Discussion, the authors state: “… rabbit (*Oryctolagus cuniculus*) mtDNA have repeated 153 bp motifs in the vicinity of the replication origin of H strand…”. Would slip-strand mispairing not fully explain this pattern? This activity is quite common in animal mtDNA.*

Thanks for pointing this out. According to the literature, slipped-strand mispairing typically occurs in repeated motifs of 1-10 bases. Therefore, it seems an unlikely explanation for deletion or addition of the longer repeats present in *D. melanogaster* mitochondrial regulatory region (∼350bp or 450bp), or the 153 bp repeats in rabbit. Nonetheless, we have changed our wording in the Discussion to make it clear that recombination is a possible explanation, not that it is the known explanation, for the shifts in repeat number.

*In the same subsection, the authors state that: “recombination among sibling molecules may be prevalent”. Why claim recombination is prevalent? The data in the Results show the opposite*.

The sentence reads, “Since recombination between sibling molecules *may* be prevalent…”. This is not a claim. This is a possibility, and it follows form the preceding paragraph. It also is restricted. It suggests that recombination might be prevalent between *sibling* molecules. Sibling molecules are the products of one replication event and are necessarily, at least transiently, co-localized, and hence not subject to all of the other barriers that might hinder interaction and recombination between two independent genomes. Consequently, there is no contradiction data from the Results, which refer to recombination between independent genomes in a heteroplasmic line. In any case, out of fear that this was not made clear by the text, we have entirely revised the presentation of these issues and left out use of the word prevalent.

*In your manuscript, you suggest that: “the main role of recombination in mitochondria is homology dependent DNA repair, which may be especially important in light of high level of DNA damage inflicted on the mtDNA by its oxidative environment”. There is growing skepticism regarding the amount of ROS damage that mtDNA experiences (example: PLoS Genet. 2014 Feb; 10(2): e1003974)*.

True, and the skepticism appears to be founded on good data. We have re-written the discussion without such a statement.

Please clarify the following note (in the subsection “Recombination upon introduction of a single DSB”): “genome was completely eliminated after 18 generations (not shown)”. Why “data not shown”? Why not show data?

Because the data have been shown as a supplementary figure in Hill, Chen, and Xu, 2014, which was also referenced. To clarify this, we omitted the “data not shown” statement and simply referred to reference.

Figure 1*: The size of the band in*
Figure 1
*looks bigger than 21.2kb. The literature (and this paper) indicate* that D. melanogaster *mtDNA is ∼19kb. Clarify this discrepancy.* D. melanogaster *mtDNA size does vary, but >21.2 kb is an unusually large size*.

Our concern in this experiment was the presence or absence of the small doubly cut band. To optimize detection we ran a gel (1.2%) more suitable for detection of a 1.6 kb band than a 19+ kb band. Additionally, we loaded a large amount of sample extracted from whole flies. Under the circumstances, we did not consider the gel an accurate way to assess the size of the genome. On the other hand, we have the full-length sequence of this genome and this sequence is deposited in conjunction with this paper. The genome is 19,542bp (Figure 2).

Figure 2*: Where are the other digestion products on the gel. There should be a ∼3kb band produced by this digest. It looks like the probe does not overlap this smaller band. It would be reassuring that all products from the digest are demonstrated on the Southern blot. Can these blots be re-probed with another probe? A PCR product for the targeted region to the right of the Xhol site would confirm this*.

Indeed, as indicated in the schematic accompanying the figure, the probe is limited to a single diagnositic band, and that is why the other bands are not visible. We disagree with the idea that it would be reassuring to see all the bands. It would be confusing. The point of this gel is not to define the distinctions in the genomes. The fully sequenced genomes do that (Figure 2). The point of the blot was to illustrate the changes in the levels of the different genomes in successive generations using a diagnostic signal and the probe was designed to provide that.

*Discussion: Can some estimates be made about the frequency of exchange in the heteroplasmic cultures? A big issue in this new finding is just how frequent this is in nature. The Discussion does a good job of clarifying that recombination will not happen without paternal leakage, mtDNAs being of different haplotypes, and being in the same organelle or ‘nucleoid’ of mtDNAs. There are two issues of interest and impact here. First, population geneticists might want to know: how much recombination is enough to purge deleterious mutation accumulation? Second, molecular geneticists will want to know how often these occur for disease issues or for constructing novel genotypes for experimental work. This frequency estimate is a rough guess, but that would increase the impact*.

We are conflicted about this request to estimate a frequency of recombination. While we are in better position than a reader to assess this, we cannot make a realistic estimate of frequency of a molecular event. Additionally, we strongly suspect that recombination is regulated and genetically modulated, and that a single estimate might be more misleading than informative. We can of course note the frequency with which females are fertile or fraction of vials yielding progeny when subjected to selection and we do report this. We also have noted what we feel is an enormous difference in frequency in the presence and absence of restriction enzyme cutting, perhaps an indication of range of regulation.

Here we list the problems in extrapolating to a molecular estimate:

1) We do not know the efficiency with which a recombination event can result in escape of selection: If recombination produces a single resistant (or functional) genome in the germline, what is the likelihood that we will recover it? We don’t know how fast the mitochondrial genomes were cut and then disappear, how long the germline cells can survive with mostly disrupted mtDNA, and how many copies of a recombinant genome is required in order to make the germline cell viable and capable of subsequent development. Maybe one recombinant genome from one successful recombination event is sufficient to survive the selection, but it has to be generated early during oogenesis, so it has the chance to replicate itself and then repopulate the whole egg, which often contains ∼10 million copies of mtDNA. Since most of the progeny we recovered are homoplasmic, it appears that only a single event was recovered, but for every event recovered perhaps there were many thousands that failed to make it.

2) Even if we knew the efficiency with which a recombinant could rescue, we need a denominator to turn the frequency of fertile females into a more meaningful frequency of recombination than the one we have already reported, and we do not know what this denominator is. We can estimate the number of germline stem cells (about 64 or perhaps twice of this if we include the first wave of direct developing stem cells), but we actually don’t know if recombination occurs in these cells or their progenitors. So we cannot accurately report a frequency on a per cell basis. Furthermore, we do not know the number of mitochondria in a cell, so we cannot change this to recombination frequency per genome.

3) Different selections are likely to result in different frequencies of recovered recombinants. The survivors of restriction enzyme selection are homoplasmic reflecting a strong selection against the starting genomes that acts to completion in a single generation with little opportunity for a slow amplification of the resistant genomes. In contrast, the temperature sensitive selection “against” the mutant allele of *mt:CoI* is actually a selection for the function of the wild-type allele and even a low abundance of functional recombinant genome can rescue survival of heteroplasmic line and gradually accumulate over multiple generations.

As a result of these uncertainties, we feel that we cannot offer a frequency. We do, however, point out here that when we induced complementary DSBs, which gave an immediate and strong selection for homoplasmic recombinant progeny, most of females had high fertility. This means that many of the germline stem cells had recombinant genomes. This requires that many recombination events occur in most females. Furthermore, preliminary data suggests that rescue of fertility by having ∼2.5% of resistant genome upon cutting, is inefficient, meaning that many recombinant genomes are required for in each stem cell to produce progeny. Given that this was in response to DSBs, we suggest that it is an indication that recombination pathways will efficiently repair DSBs. Indeed, it is likely that the recombinants observed represent a small fraction of amount of homologous repair, since the sibling genomes will presumably be the more available partners for repair but will not contribute restriction resistant genomes.

At the other extreme, when the no restriction cutting was used in the selection (in the combination of ATP6[1] and the temperature sensitive genome), recombinants were observed in the minority of vials (3 out of 51 vials), each with a small population of females (about 40), and the recombinants appeared only after several generations. While this still represents a reasonably high likelihood of observing a recombinant in a vial, we suspect that the frequency of underlying event is considerably more than two orders of magnitude less frequent than that observed in the double restriction experiment.

We would also like to note that if one mixed two genetically marked populations of *E. coli* or of the yeast *S. cerevisiae* (which in the wild is homothallic), one is unlikely to see any significant recombination, and that the intensive use of these organisms for genetics followed the discovery of specialized approaches to manipulate them to reveal their innate capacities for recombination. We now know that mitochondria can recombine and hopefully we can uncover the factors that modulate their capacity to do so.